# RNA Sequencing-Based Identification of Ganglioside GD2-Positive Cancer Phenotype

**DOI:** 10.3390/biomedicines8060142

**Published:** 2020-05-30

**Authors:** Maxim Sorokin, Irina Kholodenko, Daniel Kalinovsky, Tatyana Shamanskaya, Igor Doronin, Dmitry Konovalov, Aleksei Mironov, Denis Kuzmin, Daniil Nikitin, Sergey Deyev, Anton Buzdin, Roman Kholodenko

**Affiliations:** 1Shemyakin-Ovchinnikov Institute of Bioorganic Chemistry, Russian Academy of Sciences, 16/10, Miklukho- Maklaya St., 117997 Moscow, Russia; sorokin@oncobox.com (M.S.); dcalinovschi@yahoo.com (D.K.); doroninii@gmail.com (I.D.); nikitin@oncobox.com (D.N.); deyev@mail.ibch.ru (S.D.); buzdin@oncobox.com (A.B.); 2Sechenov First Moscow State Medical University, 8-2, Trubetskaya St., 119992 Moscow, Russia; 3Omicsway Corp., 340 S Lemon Ave, 6040, Walnut, CA 91789, USA; 4Orekhovich Institute of Biomedical Chemistry, 10, Pogodinskaya St., 119121 Moscow, Russia; irkhol@yandex.ru; 5D. Rogachev Federal Research Center of Pediatric Hematology, Oncology and Immunology, 1, Samory Mashela St., 117997 Moscow, Russia; shamanskaya.tatyana@gmail.com (T.S.); dmk_nadf@mail.ru (D.K.); 6Real Target LLC, 108841 Moscow, Russia; 7Skolkovo Institute of Science and Technology, 3, Nobelya St., 121205 Moscow, Russia; magmir71@gmail.com; 8Moscow Institute of Physics and Technology (National Research University), 141700 Moscow, Russia; kuzmin.dv@mipt.ru; 9Oncobox ltd., 121205 Moscow, Russia

**Keywords:** ganglioside GD2, GD2-positive tumors, neuroblastoma, glioma, immunotherapy, molecular diagnostics, NGS, ganglioside biosynthesis, targeted therapy, RNA sequencing

## Abstract

The tumor-associated ganglioside GD2 represents an attractive target for cancer immunotherapy. GD2-positive tumors are more responsive to such targeted therapy, and new methods are needed for the screening of GD2 molecular tumor phenotypes. In this work, we built a gene expression-based binary classifier predicting the GD2-positive tumor phenotypes. To this end, we compared RNA sequencing data from human tumor biopsy material from experimental samples and public databases as well as from GD2-positive and GD2-negative cancer cell lines, for expression levels of genes encoding enzymes involved in ganglioside biosynthesis. We identified a 2-gene expression signature combining ganglioside synthase genes *ST8SIA1* and *B4GALNT1* that serves as a more efficient predictor of GD2-positive phenotype (Matthews Correlation Coefficient (MCC) 0.32, 0.88, and 0.98 in three independent comparisons) compared to the individual ganglioside biosynthesis genes (MCC 0.02–0.32, 0.1–0.75, and 0.04–1 for the same independent comparisons). No individual gene showed a higher MCC score than the expression signature MCC score in two or more comparisons. Our diagnostic approach can hopefully be applied for pan-cancer prediction of GD2 phenotypes using gene expression data.

## 1. Introduction

Ganglioside GD2 is a well-established target for cancer immunotherapy. This glycosphingolipid is abundant on the surface of tumor cells but has restricted expression on healthy cells. GD2 is a functionally important molecule that has complex biological functions in tumor cells: it contributes to cellular adhesion [1] and mediates cytotoxic signals [2,3]. Populations of GD2-positive tumor cells are often characterized by increased proliferation and invasiveness [4,5]. Biosynthesis of GD2 is the result of complex biochemical reactions, it is synthesized from simpler ganglioside molecules under the action of several enzymes: galactosyl and sialyl transferases [6]. However, GD2 is not necessarily a terminal reaction product, but can also serve as intermediate and substrate in the synthesis of gangliosides with higher molecular mass [7]. Therefore, the activities and expression levels of different biosynthetic enzymes can either positively or negatively affect the GD2 content, thereby resulting in a GD2-positive or negative cell phenotype.

Combination therapy with monoclonal GD2-specific antibodies is now a standard of care for high-risk neuroblastoma patients [8]. While almost all neuroblastomas (≥96%) are thought to be GD2-positive, they exhibit considerable heterogeneity in the expression levels of GD2 in tumor cells [9,10]. Therapy efficiency with GD2-specific antibodies is directly dependent on the GD2 expression level, as reflected by a significantly lower probability of relapse following therapy in GD2-overexpressing tumors [10].

In addition to neuroblastoma, GD2 is also overexpressed in other tumors, such as melanoma [11], gliomas [12,13], medulloblastoma [14], different sarcomas [15,16], retinoblastoma [17], small cell lung cancer [18,19], bladder cancer [20], colorectal cancer [21], and several types of breast cancer [4,22,23]. Several clinical trials of anti-GD2 therapies in these tumors are currently underway [24].

However, the aforementioned tumors have highly variable levels of GD2 expression and, therefore, require accurate diagnostic methods for the adequate application of the GD2-targeted therapies. At present, immunohistochemistry (IHC) is the main instrument of detecting GD2 in tumor samples [9,10]. IHC staining is widely used for screening of tumor biomarkers, yet it has some limitations. The results of the staining are largely dependent on the quality of the biopsy material, and there are no universal quality controls, which sometimes can lead to inaccuracy or even erroneous conclusions [25]. At the same time, IHC analyses of GD2 are frequently problematic because gangliosides are considered unusually small antigens for accurate detection with antibodies [26]. In turn, PCR approaches cannot be directly used for GD2 screening since it is a non-protein molecule.

We hypothesized that functional balance of the enzymes involved in the biosynthesis of gangliosides may serve as a predictor of cell surface GD2 expression. In this study, we built a gene expression-based binary classifier predicting GD2-positive tumor phenotypes that reflects molecular pathway-based gene signatures many of which were previously shown to be effective in cancer diagnostics [27,28,29,30,31].

We compared RNA sequencing data from human tumor biopsy material from experimental samples and public databases as well as from GD2-positive and GD2-negative cancer cell lines, for expression levels of genes encoding enzymes involved in ganglioside biosynthesis. We identified a 2-gene expression signature combining ganglioside synthase genes *ST8SIA1* and *B4GALNT1* that serves as a more efficient predictor of GD2-positive phenotype (MCC 0.32, 0.88, and 0.98 in three independent comparisons) compared to the individual ganglioside biosynthesis genes (MCC 0.02–0.32, 0.1–0.75, and 0.04–1 for the same independent comparisons). No individual gene showed a higher MCC score than the expression signature MCC score in two or more comparisons. Our diagnostic approach can hopefully be applied for pan-cancer prediction of GD2 phenotype for the adequate application of GD2-directed therapies.

## 2. Materials and Methods

### 2.1. Cell Lines and Flow Cytometry

Human glioblastoma cell line T98G, human glioblastoma astrocytoma cell line U-373, human osteosarcoma cell line HOS, and human osteosarcoma cell line U2OS were cultured in DMEM; human neuroblastoma cell line IMR-32 and human neuroblastoma cell line SH-SY5Y were cultured in EMEM medium. All culture media were supplemented with 10% heat-inactivated fetal bovine serum (FBS), 2 mM L-glutamine, 100 μg/mL penicillin, and 100 U/mL of streptomycin (all—Thermo Fisher Scientific, Waltham, MA, USA).

Staining of cells with AF488-labelled GD2-specific antibodies 14G2a (Santa Cruz, Dallas, TX, USA) was performed as described previously [32]. In brief, cells were detached from the culture plates by trypsinization, incubated with AF488-labelled antibodies 14G2a (1 μg per 10^6^ cells) for 1 h, and then washed twice in PBS supplemented with 1% FBS and 0.02% sodium azide. All procedures were performed at 4 °C. The samples were immediately analyzed using EPICS Coulter XL-MCL flow cytometer (Beckman Coulter, Porterville, CA, USA). In each sample at least 5000 events were collected. For all samples, the analysis was performed in triplicate. The relative fluorescence intensity (RFI) of GD2 expression in each cell line was calculated as the ratio of specific fluorescence of cell staining with AF488-labelled antibodies 14G2a and autofluorescence of control unstained cells. The data were analyzed using WinMDI software.

### 2.2. Biosamples

The biospecimens used in the present study were provided by the Dmitry Rogachev Federal Research Center of Pediatric Hematology, Oncology, and Immunology. All derived samples were obtained with informed consent under institutional review board-approved protocols. Samples were stored in formalin-fixed paraffin-embedded (FFPE) tissue block at room temperature. We obtained tissue specimens from 3 patients (4, 5 and 9 years old) with high-risk neuroblastoma. This study was performed under a protocol approved by the Institutional Review Board (IRB) at Clinical Center Vitamed, Moscow, Russia (protocol date 16.10.17). Patients provided written informed consent to participate in this study.

### 2.3. Library Preparation and RNA Sequencing

RNA extraction. Cell line samples were stabilized in RNAlater (Qiagen, GmbH, Hilden, Germany) and stored at room temperature. RNA extraction was performed immediately before the preparation of sequencing libraries using QIAGEN RNeasy Kit (Qiagen) or Direct-zol RNA MiniPrep (Zymo Research, Irvine, CA, USA), followed by an additional purification step by TRI Reagent (MRC, Cincinnati, OH, USA) for cell lines in RNAlater and RecoverAll Total Nucleic Acid Isolation Kit (Invitrogen, Waltham, MA, USA) for FFPE, according to the manufacturers’ protocols. RNA was quantified using Nanodrop (Thermo Fisher Scientific, USA), ethanol-precipitated, and stored in liquid nitrogen until sequencing.

Library preparation. RNA Integrity Number (RIN) was measured using Agilent 2100 bioanalyzer (Agilent, Santa Clara, CA, USA). Agilent RNA 6000 Nano or Qubit RNA Assay (Thermo Fisher Scientific) kits were used to measure RNA concentration. KAPA RNA Hyper with RiboErase Kit (KAPA Biosystems, Wilmington, MA, USA) was used for further depletion of ribosomal RNA and library preparation. Different adaptors were used for multiplexing samples in one sequencing run. Library concentration and quality were measured using Qubit dsDNA HS Assay Kit (Thermo Fisher Scientific) and Agilent TapeStation system (Agilent). Single-end RNA sequencing was performed using Illumina HiSeq 3000 system (Illumina, San Diego, CA USA), 50 bp read length, for approx. 30 million raw reads per sample. Data quality check was conducted using Illumina SAV. De-multiplexing was performed with Illumina Bcl2fastq2 v 2.17 software.

Processing of RNA sequencing data. RNA sequencing FASTQ files were processed by STAR aligner in ‘GeneCounts’ mode with the Ensembl human transcriptome annotation (Build version GRCh38 and transcript annotation GRCh38.89). Ensembl gene IDs were converted to HGNC gene symbols using Complete HGNC dataset [33]. In total, expression levels were established for 36,596 annotated genes with the corresponding HGNC identifiers.

The sequencing data generated in this study are publicly available via Gene Expression Omnibus database (accession ID GSE92742).

### 2.4. Extraction and Processing of Publicly Available Data

RNA sequencing FASTQ files were downloaded directly from TCGA [34], GTEx [35], and TARGET [36] repositories. FASTQ files of ANTE database, gastric cancer, thyroid cancer, glioblastoma and breast and lung cancer were downloaded from NCBI Sequencing Read Archive using the accessions SRP163252, PRJNA562149, PRJNA574551, PRJNA590641, PRJNA565016 and PRJNA578290, respectively.

RNA sequencing FASTQ files for all publicly available samples analyzed in this study were processed by STAR aligner in ‘GeneCounts’ mode with the Ensembl human transcriptome annotation (Build version GRCh38 and transcript annotation GRCh38.89). Ensembl gene IDs were converted to HGNC gene symbols using Complete HGNC dataset [33]. In total, expression levels were established for 36,596 annotated genes with the corresponding HGNC identifiers.

### 2.5. Data Extraction Normalization

Read counts per gene for six cell lines were normalized using DESeq2 package (version 1.26.0) in R environment. Normalized gene counts were used as gene expression values.

We extracted RNA sequencing profiles and corresponding clinical annotations from TCGA [34], GTEx [35], and TARGET [36] databases. Additionally, we used ANTE (Oncobox Atlas of Normal Tissue Expression) database [37] combined with the expression datasets for different cancer types, obtained using the same equipment and reagents: gastric cancer [38], thyroid cancer (NCBI Sequencing Read Archive: PRJNA574551), glioblastoma (PRJNA590641), and breast and lung cancer (PRJNA565016 and PRJNA578290). We also took three tissue specimens from high-risk neuroblastoma patients and analyzed them jointly with the ANTE database of healthy tissues, profiled using the same reagents and equipment.

Totally, we analyzed 22,078 cancer and normal tissue samples, 9079 of which corresponded to TCGA, 11,681 to GTEx, 1014 to TARGET, and 304 to ANTE and compatible experimental samples (Appendix A). Read counts of all publicly available cancer and normal samples were assembled into one table consisting of 22,078 samples and jointly normalized using DESeq2 package (version 1.26.0). Similarly, normalized gene counts were used as gene expression values.

Scatterplots, violin plots, and distributions were drawn using ggplot2 package (version 3.3.0) in R environment. Heatmap drawing and cluster analysis of genes and the 2-gene expression signature were done in the space of log10-transformed DESeq2-normalized gene counts using ward.d2 method implemented in heatmap.2 function of gplots package (version 3.0.3) in R environment.

### 2.6. Statistical Analysis

In order to measure statistical significance between two datasets in each comparison we used Matthew’s Correlation Coefficient (MCC) score [39]. We calculated MCC score and *p*-values using mcc function [40] in PharmacoGx package (version 1.17.1) in R environment. Calculation of MCC score was executed with the assumption of equal type I and type II errors.

### 2.7. Gene Ontology and Random Testing of 2-Gene Signatures

We extracted all experimentally confirmed human genes with Gene Ontology term “lipid metabolic process” (GO:0006629) [41]. In total, we selected 520 annotated genes with the corresponding HGNC identifiers. For each prediction of GD2-positive cancer phenotype ((i) TNBC and non-TNBC samples from TCGA, (ii) brain tissues and all non-nerve tissue derived samples from GTEx dataset, and (iii) putatively GD2-positive neuroblastoma samples and putatively GD2-negative leukemia samples from TARGET dataset) we calculated gene signature scores for 1000 randomly chosen pairs of genes associated with lipid metabolism. Random selection was performed using the basic function “sample” in R environment. Gene signature scores were calculated as the sum of decimal logarithms of normalized expression levels of two genes. For each of these randomly generated gene signatures we calculated the MCC score for discrimination of GD2-positive and GD2-negative subsets using the assumption of equal type I and type II errors.

## 3. Results

### 3.1. Characterization of Cancer Cell Lines by GD2 Expression

Initially, we characterized human cancer cell lines of neuroblastoma, glioma, and sarcoma origin by surface expression of GD2 using flow cytometry and staining with GD2-specific antibodies. The cells of these types of cancer cell lines expressed ganglioside GD2 and therefore could be used to compare the expression level of GD2 with the expression of ganglioside biosynthesis enzyme genes obtained by RNA sequencing. To establish the GD2-positive cell phenotype, monoclonal antibodies 14G2a were used that are strictly specific for ganglioside GD2 and do not manifest cross-reactivity with other gangliosides, which was previously confirmed by ELISA on pure gangliosides and TLC data on ganglioside composition of the cell lines [2].

We investigated two human neuroblastoma cell lines IMR-32 and SH-SY5Y, human glioblastoma/astrocytoma cell lines T98G and U-373, and two human sarcoma cell lines HOS and U2OS. Every experiment was done in at least three replicates, representative results are shown on Figure 1.

Cell lines were categorized according to RFI values as GD2-overexpressing (GD2^++^; RFI >10), GD2-positive (GD2^+^; 1.5 < RFI < 10), or GD2-negative (GD2^−^; RFI < 1.5). Two of the cell lines investigated (glioblastoma T98G and neuroblastoma IMR-32) were classified as GD2^++^. Three cell lines (osteosarcoma U2OS, neuroblastoma SH-SY5Y, and astrocytoma U-373) were classified as GD2^+^. Finally, osteosarcoma HOS was classified as a GD2^−^ cell line (Figure 1).

### 3.2. Association between Expression of Ganglioside Biosynthesis Enzymes and GD2-Positive Phenotype

Gangliosides and other complex glycosphingolipids are synthesized in human cells as a result of the successive addition of sugar residues to glycosylceramide. Galactosyl and sialyl transferases, which are the main enzymes involved in these reactions, use different substrates with a similar structure and can synthesize several gangliosides [7,31]. The most important known enzymes in ganglioside biosynthesis are ST3GAL5, ST8SIA1, ST8SIA5, B3GALT4, B4GALNT1, and B4GALT6 (list of abbreviations in Appendix A), while the biosynthesis of GD2 is directly carried out by ST8SIA1 and B4GALNT1 enzymes (Figure 2).

We profiled gene expression in the six cell lines under investigation by RNA sequencing (for two GD2^++^, three GD2^+^, and one GD2^-^ cell lines). We then normalized raw sequencing read counts for these cell lines and plotted gene expression values for each of the six genes related to GD2 synthesis: *ST3GAL5*, *ST8SIA1*, *ST8SIA5*, *B3GALT4*, *B4GALNT1*, and *B4GALT6* (Figure 3). For each of these genes, the expression levels were compared between the cell lines with different GD2 status. We also constructed a preliminary gene signature consisting of two most GD2-proximal enzymes, GD2 and GD3 synthases that catalyze synthesis of GD2 and its immediate precursor GD3 (genes *B4GALNT1* and *ST8SIA1*, respectively; Figure 3). The signature consisting of genes *B4GALNT1* and *ST8SIA1* was calculated as the sum of decimal logarithms of normalized expression levels of *B4GALNT1* and *ST8SIA1*. Both the 2-gene signature and expressions of the most “GD2-proximal” genes *ST8SIA1* and *B4GALNT1* were congruent with the functional GD2-statuses of the analyzed cell lines (Figure 3).

In order to compare the *ST8SIA1* + *B4GALNT1* gene signature to other combinations, we created all possible gene pairs using the aforementioned six genes related to GD2 synthesis. For each pair gene signature scores were calculated as the sum of log10-transformed gene expression values of specified genes for cell lines of three GD2 phenotypic groups (GD2++, GD2+, and GD2-). Expression values for genes coding enzymes that catalyze downstream reactions that decrease GD2 concentration (*ST8SIA5* and *B3GALT4*) were taken with a minus sign. Among the 15 gene pairs tested only 5 gene signatures (*ST8SIA1+B4GALT6*, *ST8SIA1+B4GALNT1*, *B4GALT6+B4GALNT1*, *B4GALT6-B3GALT4* and *B4GALNT1−B3GALT*) were congruent with the functional GD2-statuses of the tested cell lines, i.e., all GD2-positive samples showed gene expression scores higher than the GD2-negative one (Appendix A). The *ST8SIA1* and *B4GALNT1* pair was selected for further studies from those 5 gene pairs because it also discriminates GD++ from GD+ phenotype (Appendix A).

### 3.3. Trends in Ganglioside Biosynthesis Enzyme Expressions in Public Gene Expression Databases

We then analyzed expressions of the ganglioside synthesis genes in different cancer types using public gene expression datasets. We extracted RNA sequencing profiles and corresponding clinical annotations from TCGA [34], GTEx [35], and TARGET [36] databases, as well as ANTE (Oncobox Atlas of Normal Tissue Expression) database [37] combined with the expression datasets for different cancer types, obtained using the same equipment and reagents: gastric cancer [38], thyroid cancer (NCBI Sequencing Read Archive: PRJNA574551), glioblastoma (PRJNA590641), and breast and lung cancer (PRJNA565016 and PRJNA578290, respectively). We also took three tissue specimens from high-risk neuroblastoma patients. The individual samples were classified by tumor types according to the annotations extracted from the corresponding data repositories. For the TCGA breast cancer dataset, we separately took triple-negative breast cancer (TNBC) and other breast cancer molecular subtypes since TNBC is associated with GD2-positive phenotype unlike other subtypes which predominantly have GD2-negative phenotype [22].

Raw gene-mapped read counts extracted from the databases were collectively normalized, and the expression values for the ganglioside biosynthesis genes as well as the gene signature scores (sum of log10-transformed expressions of *ST8SIA1* and *B4GALNT1*) were plotted on heatmaps (Figure 4). Sample types for each database (TCGA, GTEx, ANTE plus platform-compatible cancer samples, and TARGET, respectively) were sorted in descending order of the gene signature score. We also did cluster analysis in order to assess whether the gene signature proposed confers novel patterns of gene expression that are irreducible to individual genes [18].

For each database, the 2-gene signature showed separate first-order clustering thus also demonstrating that it is an independent biomarker with respect to individual genes. Moreover, sorting by the 2-gene signature score in descending order placed normal and cancerous neural tissues as well as other GD2-positive cancers and tissues on the top of all heatmaps. For example, in the case of the TCGA dataset (Figure 4A), the highest scores were demonstrated by glioblastoma and low-grade glioma, whereas TNBC samples had a higher 2-gene signature score compared with non-TNBC samples. The same pattern was observed for the ANTE dataset plus platform-compatible experimental neuroblastoma samples (Figure 4C). Furthermore, brain tissues had maximal scores in both GTEx (Figure 4B) and ANTE (Figure 4C) datasets, while neuroblastoma and glioblastoma were on the second and third positions respectively in ANTE (Figure 4C). Similarly, neuroblastoma showed the highest 2-gene signature score in the TARGET dataset (Figure 4D).

In order to examine whether the proposed 2-gene signature can serve as a better GD2-predictive biomarker compared to individual genes involved in ganglioside biosynthesis, we analyzed the following pairs from GD2-positive and GD2-negative datasets: (i) triple-negative breast cancer (TNBC) and non-TNBC samples from TCGA (84 vs. 897 samples, respectively), (ii) brain tissues and all non-nerve tissue derived samples from GTEx dataset (1671 vs. 9820 samples, respectively), and (iii) putatively GD2-positive neuroblastoma [9,10] samples and putatively GD2-negative leukemia [43,44] samples from TARGET dataset (165 vs. 531 samples, respectively). Different types of breast cancer were used because it was recently reported that the presence of GD2 is characteristic of the triple-negative breast cancer [22]. Normal brain tissue was compared to other non-nerve tissues from GTEx project, because GD2 is highly expressed on the CNS neurons [45,46] where ganglioside biosynthesis is activated. For each pair, we assessed the predictive power of each ganglioside biosynthesis gene and of the 2-gene signature to separate the GD2-positive and GD2-negative samples. The datasets for each pair were highly dissimilar in sample size, and we thus assessed the predictive power using Matthews Correlation Coefficient (MCC) (Figure 5).

In all but two cases, the 2-gene signature showed higher MCC values than the individual genes. Specifically, in the case of TNBC vs. non-TNBC comparison (Figure 5A) the 2-gene signature gave MCC = 0.32, which was equal to the MCC for *ST8SIA1* and higher than the MCC for other genes. Furthermore, the comparison of healthy brain vs. non-neural tissues (Figure 5B) resulted in the highest MCC value (0.88) for the 2-gene signature. Finally, neuroblastoma and leukemia samples were compared, the 2-gene signature had an MCC value slightly lower than *B4GALNT1* (0.98 and 1, respectively) but higher than the other genes. We, therefore, concluded that individual genes *ST8SIA1* and *B4GALNT1* can serve as effective biomarkers in particular cases, but the expression signature based on both of them was robust in all the comparisons under analysis.

In order to test whether the observed difference in publicly available RNAseq datasets is specific to GD2 and is not likely to be attributed to other lipids we compared the “ST8SIA1 + B4GALNT1” signature with random gene pairs linked with lipid metabolism. For that we extracted all experimentally confirmed human genes with Gene Ontology term “lipid metabolic process” (GO:0006629) [41]. In total 520 annotated genes with the corresponding HGNC identifiers were extracted. For each prediction of GD2-positive cancer phenotype ((i) TNBC and non-TNBC samples from TCGA, (ii) brain tissues and all non-nerve tissue derived samples from GTEx dataset, and (iii) putatively GD2-positive neuroblastoma samples and putatively GD2-negative leukemia samples from TARGET dataset) we calculated gene signature scores for 1000 randomly chosen pairs of genes connected with lipid metabolism. Gene signature scores were calculated as the sum of decimal logarithms of normalized expression levels of two genes. For each of these randomly generated gene signatures we calculated the MCC score for the separation of GD2-positive and GD2-negative subsets using the assumption of equal type I and type II errors. Then for each prediction of GD2-positive cancer in publicly available RNAseq datasets, we plotted the distribution of randomly generated MCC scores for gene signatures connected with lipid metabolism and compared this distribution to actual MCC scores calculated for the GD2 gene signature. We computed the *p*-value as a frequency of randomly generated MCC scores that were higher than the MCC score of the GD2 gene signature. For each comparison (TCGA, GTEx, and TARGET) these *p*-values were lower than 0.05 (Figure 6).

## 4. Discussion

Immunotherapeutic strategies targeting ganglioside GD2 are on the rise following the approval of the GD2-specific chimeric antibody Unituxin that demonstrates clinical efficacy in patients. A number of strategies aimed at the development of more effective molecules with reduced side effects may be employed to reinforce GD2-targeted antibody therapy. One of those is modification of the FDA-approved chimeric antibody Unituxin aimed at the development of more effective molecules with reduced side effects. The Hu14.18K322A antibody currently in clinical trials [47] that is expected to reduce immunogenicity and pain in patients, which are the main adverse effects of Unituxin, represents an example of this approach. In addition, numerous studies are underway regarding construction of various immunoconjugates and bispecific antibodies based on both full-length antibodies and their fragments [48,49]. Optimization of the mode and regimen of administration of GD2-specific antibodies can also significantly improve the therapy outcomes. For example, prolonged intravenous infusion of Unituxin reduces side effects characteristic of the standard short-term infusion procedure [50]. Yet another strategy consists in the selection of combination therapies of anti-GD2 antibodies with chemotherapy and cytokines [51]. To this end, RNA sequencing and bioinformatic approaches can be applicable [52]. Since direct induction of cell death [53] through different signaling pathways [3,54,55] is one of the mechanisms of antitumor activity of GD2-specific antibodies, the given approach can be adapted for GD2-targeted combination therapies [56].

A different application relying on RNA sequencing data is identification of the GD2 tumor phenotype. Potentially, this characteristic can determine the need to administer GD2-targeted therapy and predict its efficiency. The GD2-positive phenotype depends on the activities of the ganglioside biosynthesis enzymes. The present study as well as several others [4,57] demonstrates that GD3 synthase (*ST8SIA1*) and GD2 synthase (*B4GALNT1*) genes play the major roles in the expression of GD2. Specifically, we showed that the level of gene expression of these enzymes demarcates GD2-positive from GD2-negative samples in experimental cell lines and neuroblastoma tissues. Moreover, the expression level of these enzymes most clearly and statistically significantly correlated with putative GD2-positive phenotype in public datasets of RNA sequencing profiles.

The combined evaluation of *ST8SIA1* and *B4GALNT1* gene expression is a better predictor of the putative GD2-positive phenotype compared to the expression of the individual genes *ST8SIA1* and *B4GALNT1*. Our data also suggest that the 2-gene expression signature of *ST8SIA1* and *B4GALNT1* can potentially represent a better biomarker than the expression levels of individual ganglioside biosynthesis genes. Further experimental evaluation of GD2 status and gene expression in tumor biopsies is needed to validate this hypothesis.

We assessed the ability of the 2-gene signature to discriminate between the GD2-positive and GD2-negative phenotypes, by using RNA sequencing data for putatively GD2-positive neuroblastoma samples [9,10] and putatively GD2-negative leukemia samples [43,44] from TARGET database. We showed that the 2-gene signature had nearly equal predictive power compared to *B4GALNT1* gene (MCC 0.98 and 1, respectively), whereas *ST8SIA1* gene showed a lower MCC score of 0.84.

It is generally accepted that GD2 is a marker of neuroectodermal tumors [58,59]. GD2 was also described as a tumor stem cell marker [60,61] primarily in the case of breast cancer [4]. It was recently reported that the presence of GD2 is characteristic of triple-negative breast cancer [22]. We evaluated gene expression data in TNBC and non-TNBC breast cancer samples using the TCGA dataset and observed higher expressions of *ST8SIA1* and *B4GALNT1* genes, and a higher score of the 2-gene signature in TNBC. Importantly, we showed that the 2-gene signature had equal predictive power compared to the gene *ST8SIA1* (MCC 0.32 in both cases), whereas *B4GALNT1* gene showed lower MCC score of 0.21. Our data are in line with the previous findings suggesting that GD2 can be a marker of TNBC [22]. Additionally, we found that pheochromocytoma, paraganglioma, esophageal cancer, and seminoma had an increased score of the 2-gene signature. To our knowledge, there is no published data regarding GD2 status in these tumors. Thus, they may represent potential candidates for assessing efficacy of anti-GD2 therapy if their predicted GD2-positive phenotype is confirmed by alternative methods.

In the human organism, GD2 is also expressed on normal body cells such as CNS neurons, melanocytes, and bone marrow mesenchymal stem cells [45,46] where ganglioside biosynthesis is activated. We compared gene expression data of brain tissue versus other normal tissues from the GTEx project, except for adrenal gland tissue since chromaffin cells have neural crest origin. Both *B4GALNT1* and *ST8SIA1* genes demonstrated high accuracy of putative GD2 phenotype prediction (MCC score of 0.7 and 0.75, respectively), while the 2-gene signature showed the highest performance with MCC of 0.88.

Taken together, these statistical comparisons show that the 2-gene signature provides the highest GD2-positive phenotype prediction power in the three cases mentioned above (TNBC vs. non-TNBC samples from TCGA, brain tissues vs. all non-nerve tissue derived samples from GTEx, and putatively GD2-positive neuroblastoma samples vs. putatively GD2-negative leukemia samples from TARGET dataset) compared to individual genes coding enzymes of ganglioside biosynthesis. Importantly, we compared the observed GD2 signature patterns in publicly available RNA-seq datasets with randomly selected genes associated with lipid metabolism. Our analysis demonstrated that the predictive power of the *B4GALNT1* + *ST8SIA1* gene signature is significantly higher than randomly selected pairs of lipid metabolic genes.

In this study, for the first time expression levels of different members of the ganglioside biosynthesis pathway were compared in various cancerous and normal tissues derived from largest public databases. We found that the 2-gene signature (*ST8SIA1* + *B4GALNT1*) displays synergistic properties in predicting putative GD2-positive phenotype. Our results may hopefully be adapted for the prediction of the GD2 phenotype in clinical specimens from cancer patients. Such a diagnostic platform may be based on qRT-PCR, NanoString assay, or targeted RNA sequencing. However, further studies with larger groups of patients are needed to validate this approach and to estimate thresholds that vary between different cancer types.

## 5. Conclusions

The primary objective of the present study was to validate the RNA sequencing-based approach for determination of the GD2 phenotype of the tumor for the adequate application of GD2-directed therapies for various types of cancer. GD2-directed therapy is gaining momentum and demonstrates clinical efficiency, but a lack of methods exists for determining the presence of ganglioside GD2 on tumor cells. We compared the expression level of different members of the ganglioside biosynthesis pathway in various cancerous and normal tissues derived from largest public databases. We found that the 2-gene signature (*ST8SIA1* + *B4GALNT1*) displays synergistic properties in predicting putative GD2-positive phenotype. Our results may hopefully be adapted for the prediction of the GD2 phenotype in clinical specimens from cancer patients. Such a diagnostic platform may be based on qRT-PCR, NanoString assay, or targeted RNA sequencing.

## Figures and Tables

**Figure 1 biomedicines-08-00142-f001:**
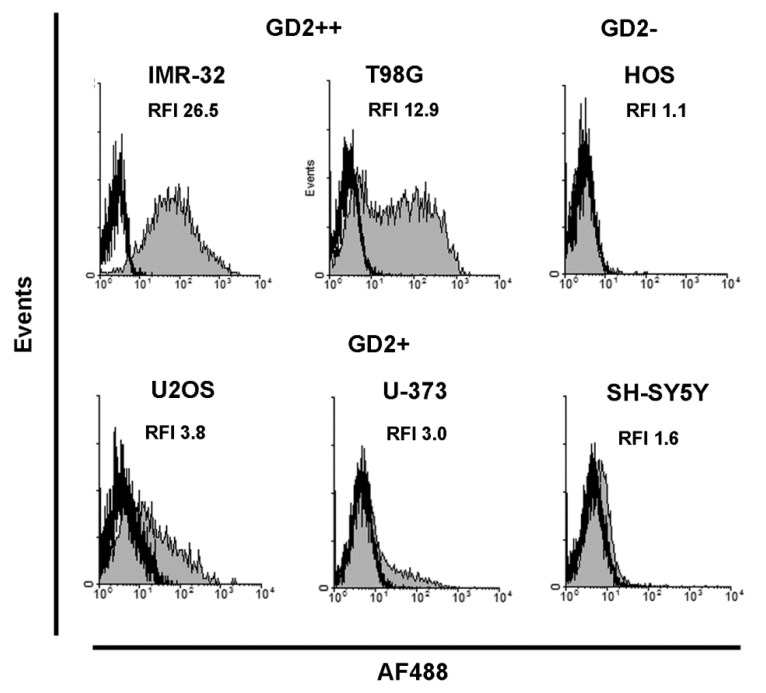
Ganglioside GD2 surface expression in cancer cell lines analyzed by flow cytometry. Gray histograms represent staining with anti-GD2 monoclonal antibodies, empty histograms represent autofluorescence of unstained cells. RFI (relative fluorescence intensity)—the ratio of specific fluorescence of cell staining with AF488-labelled antibodies 14G2a and autofluorescence of control unstained cells.

**Figure 2 biomedicines-08-00142-f002:**
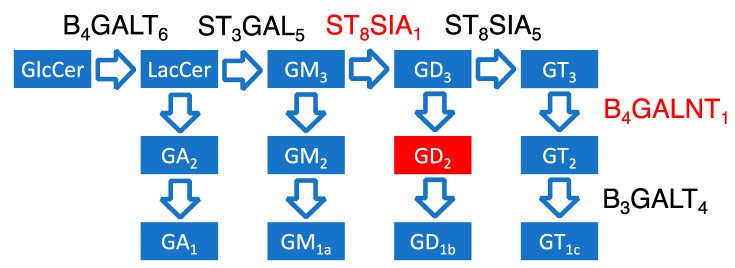
Representation of the ganglioside biosynthesis pathway and potential roles of galactosyl and sialyl transferase enzymes in GD2 synthesis. GlcCer—glucosylceramide, LacCer—lactosylceramide, B4GALT6—LacCer synthase, ST3GAL5—GM3 synthase, ST8SIA1—GD3 synthase, ST8SIA5—GT3 synthase, B4GALNT1—GD2 synthase, B3GALT4—GD1b synthase. The most GD2-proximal enzymes, GD2 synthase (B4GALNT1) and GD3 synthase (ST8SIA1) that catalyze direct biosynthetic steps leading to ganglioside GD2, are marked red. Ganglioside nomenclature used according to [42] (list of abbreviation in Appendix A).

**Figure 3 biomedicines-08-00142-f003:**
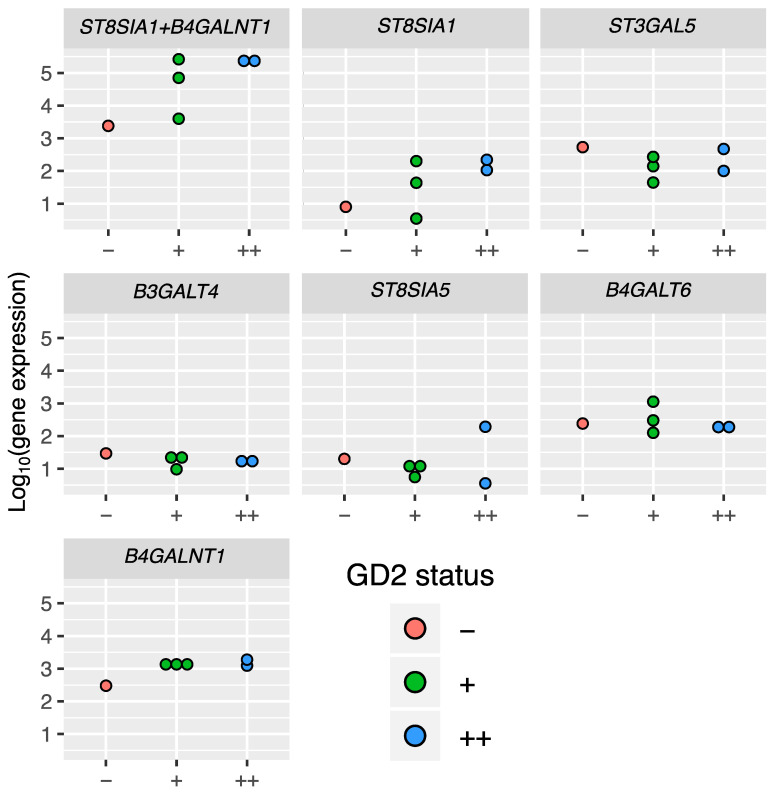
Expression of GD2 biosynthesis enzyme genes in cell lines with different GD2 status. Each panel, except *ST8SIA1* + *B4GALNT1*, corresponds to a single gene and contains log10-transformed normalized gene expression values for cell lines of three GD2 phenotypic groups (GD2^++^, GD2^+^, and GD2^−^). The panel ‘*ST8SIA1* + *B4GALNT1*’ represents gene signature scores calculated as sum of log10-transformed gene expression values of genes *ST8SIA1* and *B4GALNT1*.

**Figure 4 biomedicines-08-00142-f004:**
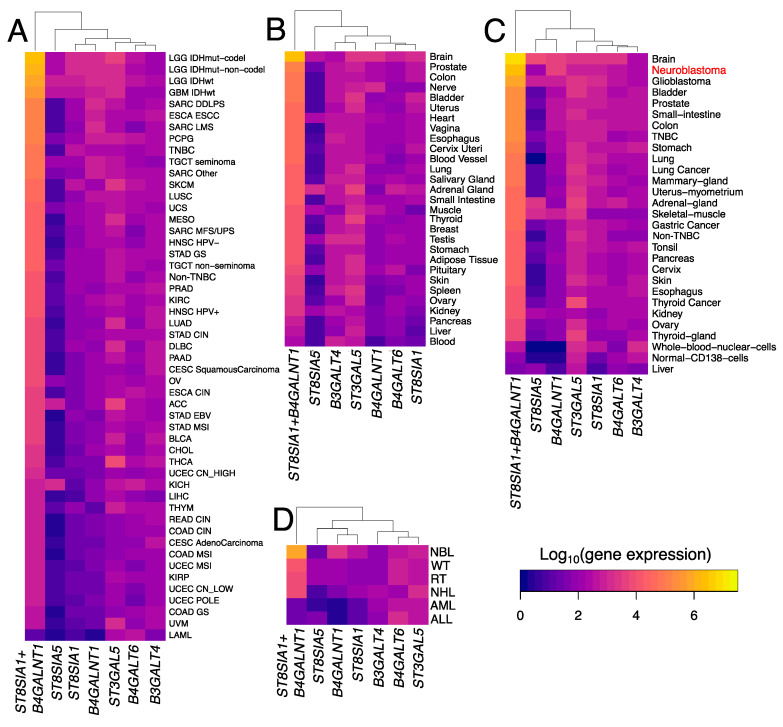
Heatmaps showing gene expression of ganglioside biosynthesis enzymes and the ‘*ST8SIA1 + B4GALNT1*’ gene signature score for cancer and healthy human tissues. (**A**) TCGA database profiles, (**B**) GTEx database profiles, (**C**) ANTE database profiles and compatible cancer samples, (**D**) TARGET database profiles. Color scale indicates log10-transformed DESeq2-normalized gene expression values. Sample types in each panel are sorted in descendent order by 2-gene gene signature score. TCGA disease codes and abbreviations: LAML: acute myeloid leukemia; ACC: adrenocortical carcinoma; TNBC: triple negative breast cancer; Non-TNBC: non-triple negative breast cancer; CESC: cervical squamous cell carcinoma and endocervical adenocarcinoma; KICH: kidney chromophobe; KIRC: kidney renal clear cell carcinoma; COAD: colorectal adenocarcinoma; SKCM: skin cutaneous melanoma; DLBC: lymphoid neoplasm diffuse large B-cell lymphoma; GBM: glioblastoma multiforme; HNSC: head and neck squamous cell carcinoma; LIHC: liver hepatocellular carcinoma; LGG: brain lower grade glioma; LUAD: lung adenocarcinoma; LUSC: lung squamous cell carcinoma; OV: ovarian serous cystadenocarcinoma; KIRP: kidney renal papillary cell carcinoma; THCA: thyroid carcinoma; STAD: stomach adenocarcinoma; PRAD: prostate adenocarcinoma; BLCA: bladder urothelial carcinoma; UCS: uterine carcinosarcoma; UCEC: uterine corpus endometrial carcinoma; ESCA: esophageal carcinoma; PCPG: pheochromocytoma and paraganglioma; PAAD: pancreatic adenocarcinoma; MESO: mesothelioma; UVM: uveal melanoma; SARC: sarcoma; CHOL: cholangiocarcinoma; TGCT: testicular germ cell tumors; THYM: thymoma; EBV: Epstein–Barr virus; DDLPS: dedifferentiated liposarcoma; LMS: leiomyosarcoma; MFS/UPS: myxofibrosarcoma/undifferentiated pleomorphic sarcoma; ESCC: esophageal squamous cell carcinoma; HPV+: human papillomavirus +; HPV-: human papillomavirus -; GS: genomically stable; CIN: chromosomal instability; MSI: microsatellite instability; CN: copy number; IDHmut-codel: IDH mutant with 1p/19q codeletion; IDHmut-non-codel: IDH mutant without 1p/19q codeletion; POLE: POLE-mutated. TARGET disease codes and abbreviations: AML: acute myeloid leukemia; ALL: acute lymphoblastic leukemia; NBL: neuroblastoma; RT: rhabdoid tumor; WT: Wilms’ tumor (nephroblastoma); NHL: non-Hodgkin lymphoma.

**Figure 5 biomedicines-08-00142-f005:**
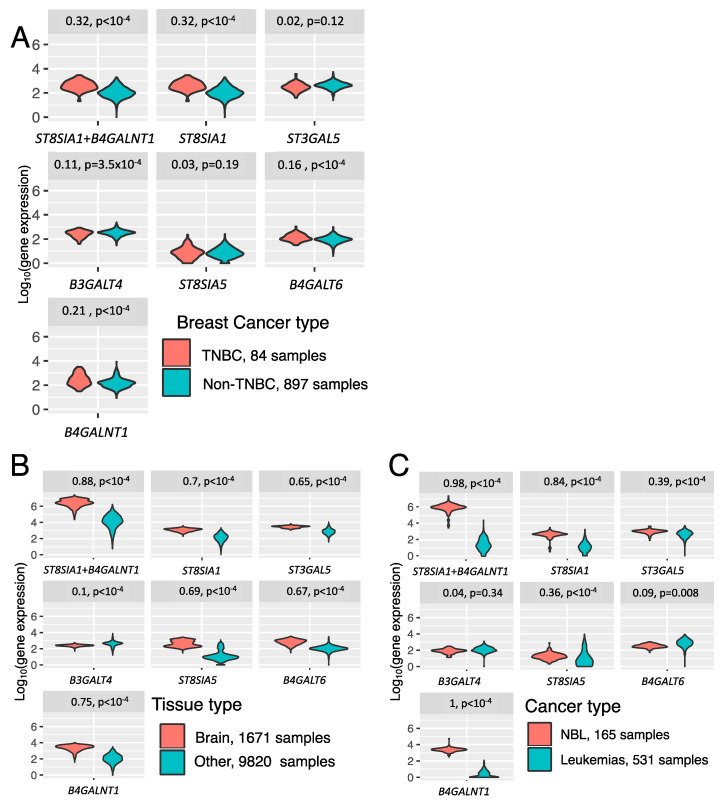
Difference in gene expression and the 2-gene signature score between putative GD2-positive (red) and putative GD2-negative (cyan) cancers and healthy tissue types. (**A**) TCGA dataset, *n* = 981 samples, (**B**) GTEx dataset, *n* = 11,491 samples, (**C**) TARGET dataset, *n* = 696 samples. Matthews Correlation Coefficient values (left) and *p*-values (right) are shown for each individual ganglioside biosynthesis gene or the 2-gene signature.

**Figure 6 biomedicines-08-00142-f006:**
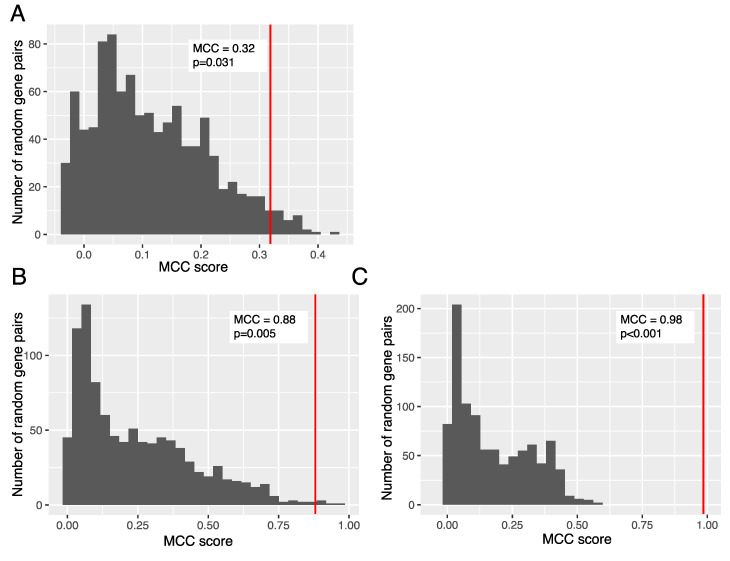
MCC score difference between the GD2 gene signature (sum of log10-transformed expressions of *ST8SIA1* and *B4GALNT1*) and 2-gene signatures consisting of randomly selected genes associated with lipid metabolism. (**A**) TNBC and non-TNBC samples from TCGA dataset, *n* = 981 samples, (**B**) brain tissues and all non-nerve tissue derived samples from GTEx dataset, *n* = 11491 samples, (**C**) putatively GD2-positive neuroblastoma samples and putatively GD2-negative leukemia samples from TARGET dataset, *n* = 696 samples.

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
