# Peer review of "RNA Sequencing-Based Identification of Ganglioside GD2-Positive Cancer Phenotype"

_biomedicines, 2020, doi:10.3390/biomedicines8060142_

Round 1
Reviewer 1 Report
This manuscript submitted by Sorokin et al investigated the expression of ganglioside GD2 in many kinds of cancers and identified a 2-gene (ST8SIA1 and B4GALNT1) expression marker for cancer prediction of GD2-positive phenotypes. This has some interesting results. However, I don’t think that the present set of data is conclusive enough to draw conclusion and that the results are not well discussed. Authors should do some additional experiments and discuss these results. Then, I would suggest extensive revision in combination with re-review for this manuscript.
Comments
- My main scientific concern is that it is not clear what marker of cancer GD2 can be in this paper-the grade of cancer, metastatic…?
- Figure 3: how the expression data “ST8SIA1+B4GALNT1” was calculated? The individual genes do not appear to differ in expression. It is hard to understand why combining the two makes a difference. You should clarify exactly what you did. And other combinations should be considered.
- You should examined the expression of other lipids and discuss these results in relation to GD2.
Author Response
Thank you very much for the detailed analysis of our manuscript and valuable suggestions. We have revised the manuscript accordingly and hope that its quality has improved. Please, find our responses to your questions and comments below. Changes in the text are highlighted with color.
Reviewer 1.
My main scientific concern is that it is not clear what marker of cancer GD2 can be in this paper-the grade of cancer, metastatic…?
The primary objective of the present study is to validate the RNA sequencing-based approach that determines the GD2 phenotype of the tumor for the adequate application of GD2-directed therapy for various types of cancer. GD2-directed therapy is gaining momentum and demonstrates clinical efficiency, but a lack of methods exists for determining the presence of ganglioside GD2 on tumor cells. We hope that the approach presented in the article contributes to solving this issue.
We have now placed additional emphasis on the main objective of the study in the Introduction section of the article. Also, for a more accurate and concise presentation of the findings of the study, we introduced the Conclusion section where the key results are explained.
It is likely that in the future the RNA sequencing-based approach will be successfully employed also to give a quantitative assessment of the level of expression of ganglioside GD2 on tumor cells, which will also allow a more detailed assessment of the diagnostic picture of the disease and predict therapy effectiveness. According to recent data, in neuroblastoma, the level of GD2 expression correlates with the stage of the disease, the level of amplification of MYCN, and a number of other clinical parameters [Balis FM et al. The ganglioside GD2 as a circulating tumor biomarker for neuroblastoma.Pediatr Blood Cancer. 2020 Jan;67(1):e28031. doi: 10.1002/pbc.28031. Epub 2019 Oct 14]. In the case of breast cancer, there is evidence of a positive relationship between the expression of GD2 and the ability to metastasize and tumorigenicity [Mansoori, M.; Roudi, R.; Abbasi, A.; Abolhasani, M.; Abdi Rad, I.; Shariftabrizi, A.; Madjd, Z. High GD2 expression defines breast cancer cells with enhanced invasiveness. Exp. Mol. Pathol. 2019, 109, 25–35.]. However, further studies with larger groups of patients are needed to validate such correlations.
Figure 3: how the expression data “ST8SIA1+B4GALNT1” was calculated? The individual genes do not appear to differ in expression. It is hard to understand why combining the two makes a difference. You should clarify exactly what you did. And other combinations should be considered.
We added a clear description how signature (“ST8SIA1+B4GALNT1”) was calculated in a passage before Fig. 3: “The signature consisting of genes B4GALNT1 and ST8SIA1 was calculated as the sum of decimal logarithms of normalized expression levels of B4GALNT1 and ST8SIA1.”
The individual genes (ST8SIA1 and B4GALNT1) differ in expression between different phenotypic groups (GD2++, GD2+, and GD2-), with an exception: one GD2+ sample has ST8SIA1 expression lower than GD2- sample. However, combining these two genes results in a biomarker which perfectly predicts GD2 phenotype in our 6 six cell lines. Due to the small group size we also tested how this gene signature predicts putative GD2+ phenotype in publicly available data. Indeed, this gene signature provides better predictions of GD2-positive phenotypes for publicly available RNAseq datasets compared to individual genes (Fig.4 and Fig.5). We appreciate the proposal to consider all other gene combinations. We added the following passage to the end of section 2.2:
“In order to compare ST8SIA1 + B4GALNT1 gene signature to other possible combinations, we created all possible gene pairs using the above six genes related to GD2 synthesis. For each pair we calculated gene signature scores as sum of log10-transformed gene expression values of specified genes for cell lines of three GD2 phenotypic groups (GD2++, GD2+, and GD2-). Expression values for genes coding enzymes that catalyze downstream reactions that decrease GD2 concentration (ST8SIA5 and B3GALT4) were taken with a minus sign. Among 15 gene pairs tested only 5 gene signatures (ST8SIA1+B4GALT6, ST8SIA1+B4GALNT1, B4GALT6+B4GALNT1, B4GALT6-B3GALT4 and B4GALNT1−B3GALT) were congruent with the functional GD2-statuses of the cell lines tested, i.e. all GD2-positive samples showed gene expression scores higher than GD2-negative one (Supplementary Figure S1). From these 5 gene pairs we selected for further studies the ST8SIA1 and B4GALNT1 pair because it also discriminates GD++ from GD+ phenotype (Supplementary Figure S1).”
You should examined the expression of other lipids and discuss these results in relation to GD2.
In general, the ganglioside composition of cells is diverse, and their functional properties are significant both in healthy and in tumor cells, which was described in detail in a number of review articles. Yet, the main direction of our research is to study the functional properties of ganglioside GD2, to develop diagnostic methods aimed at this tumor marker and molecules with potential for the treatment of GD2-positive tumors.
We introduced the reference to our earlier publication in the Results section in which the ganglioside composition of GD2-positive and GD2-negative cells by TLC was determined. Such an assessment of specific types of gangliosides justifies the use of GD2-specific antibodies for reliable determination of the GD2-positive phenotype of tumor cells.
We also performed the following bioinformatical analysis regarding other lipids and added corresponding relevant passages to address this point:
“In order to test whether the observed difference in publicly available RNAseq datasets is specific to GD2 and is not likely to be attributed to other lipids we compared “ST8SIA1 + B4GALNT1” signature with random gene pairs linked with lipid metabolism. For that we extracted all experimentally confirmed human genes with Gene Ontology term “lipid metabolic process” (GO:0006629) [43]. In total we extracted 520 annotated genes with the corresponding HGNC identifiers. For each prediction of GD2-positive cancer phenotype ((i) TNBC and non-TNBC samples from TCGA (ii) brain tissues and all non-nerve tissue derived samples from GTEx dataset and (iii) putatively GD2-positive neuroblastoma samples and putatively GD2-negative leukemia samples from TARGET dataset) we calculated gene signature scores for 1000 randomly chosen pairs of genes connected with lipid metabolism. Gene signature scores were calculated as the sum of decimal logarithms of normalized expression levels of two genes. For each of these randomly generated gene signatures we calculated MCC score for separation GD2-positive and GD2-negative subsets using the assumption of equal type I and type II errors. Then for each prediction of GD2-positive cancer in publicly available RNAseq datasets we plotted distribution of randomly generated MCC scores for gene signatures connected with lipid metabolism and compared this distribution to actual MCC scores calculated for GD2 gene signature. We computed p-value as a frequency of randomly generated MCC scores that were higher that MCC score of GD2 gene signature. For each comparison (TCGA, GTEx and TARGET) these p-values were lower than 0.05 (Figure 6).”
And
“Importantly, we compared the observed GD2 signature patterns in publicly available RNA-seq datasets with randomly selected genes associated with lipid metabolism. Our analysis demonstrated that the predictive power of the B4GALNT1 + ST8SIA1 gene signature is significantly higher than for randomly selected pairs of lipid metabolic genes.”
And
“4.7 Gene Ontology and random testing of 2-gene signatures
We extracted all experimentally confirmed human genes with Gene Ontology term “lipid metabolic process” (GO:0006629) [43]. In total we selected 520 annotated genes with the corresponding HGNC identifiers. For each prediction of GD2-positive cancer phenotype ((i) TNBC and non-TNBC samples from TCGA (ii) brain tissues and all non-nerve tissue derived samples from GTEx dataset and (iii) putatively GD2-positive neuroblastoma samples and putatively GD2-negative leukemia samples from TARGET dataset) we calculated gene signature scores for 1000 randomly chosen pairs of genes associated with lipid metabolism. Random selection was performed using the basic function “sample” in R environment. Gene signature scores were calculated as the sum of decimal logarithms of normalized expression levels of two genes. For each of these randomly generated gene signatures we calculated MCC score for discrimination of GD2-positive and GD2-negative subsets using the assumption of equal type I and type II errors.”
Reviewer 2 Report
The paper presented by M. Sorokin et al. is a very interesting and useful study concerning the cancer diagnosis based on ganglioside GD2-positive phenotype. Although the paper is detailed and contains many results the way in which the manuscript is written is not the properly one. In the Results section there are too many information regarding the methods. Also the Discussion section is not focused only on the results obtained in this study. All these facts made the paper a little confusing one. This is why I suggest to the authors to be more specific on the original results and to discuss them more deeply. I also suggest to the authors to add a Conclusions section as I think that it is important to evidence the findings of the research they performed.
Author Response
Thank you very much for the detailed analysis of our manuscript and valuable suggestions. We have revised the manuscript accordingly and hope that its quality has improved. Please, find our responses to your questions and comments below. Changes in the text are highlighted with color.
The paper presented by M. Sorokin et al. is a very interesting and useful study concerning the cancer diagnosis based on ganglioside GD2-positive phenotype. Although the paper is detailed and contains many results the way in which the manuscript is written is not the properly one. In the Results section there are too many information regarding the methods.
We strongly appreciate the assessment and the proposals of the Reviewer. A part of the data from the Results section that applies to methods was moved to the appropriate section.
Also the Discussion section is not focused only on the results obtained in this study. All these facts made the paper a little confusing one. This is why I suggest to the authors to be more specific on the original results and to discuss them more deeply.
We appreciate the proposal to be more specific on the original results and to discuss them more deeply. We added more discussion of our results in order to improve the Discussion section and to increase the overall clarity.
I also suggest to the authors to add a Conclusions section as I think that it is important to evidence the findings of the research they performed.
We fully agree with the necessity to introduce a Conclusion section. It was added to the paper.
Round 2
Reviewer 1 Report
All response is OK.